# Exploring the potential impact of the proposed UK TV and online food advertising regulations: a concept mapping study

Hannah Forde [1], Emma J Boyland [2], Peter Scarborough [3,4] Richard Smith [5], Martin White [1], Jean Adams [1]

[1]MRC Epidemiology Unit, Cambridge, UK
[2]Department of Psychology, Institute of Population Health, University of Liverpool, Liverpool, UK
[3]Nuffield Department of Population Health, University of Oxford, Oxford, UK
[4]NIHR Oxford Biomedical Research Centre, Oxford, UK
[5]Institute of Health Research, University of Exeter Medical School, Exeter, UK

**Correspondence to**
Dr Hannah Forde;
hf332@medschl.cam.ac.uk

## ABSTRACT

**Objectives** In July 2020 the UK Government announced an intention to restrict advertisements for products high in fat, salt or sugar on live broadcast, catch-up and on-demand television before 21:00 hours; and paid for online advertising. As no other jurisdiction has implemented similar regulations, there is no empirical evidence about how they might perturb the food system. To guide the regulations' implementation and evaluation, we aimed to develop a concept map to hypothesise their potential consequences for the commercial food system, health and society.

**Methods** We used adapted group concept mapping in four virtual workshops with food marketing and regulation experts across academia, civil society, government organisations, and industry (n=14), supported by Miro software. We merged concepts derived from the four workshops to develop a master map and then invited feedback from participants via email to generate a final concept map.

**Results** The concept map shows how the reactions of stakeholders to the regulations may reinforce or undermine the impact on the commercial food system, health and society. The map shows adaptations made by stakeholders that could reinforce, or undermine, positive impacts on public health. It also illustrates potential weaknesses in the design and implementation of the regulations that could result in little substantial difference to public health.

**Conclusions** Prior to the regulations' initial implementation or subsequent iterations, they could be altered to maximise the potential for reinforcing adaptations, minimise the potential for undermining adaptations and ensure they cover a wide range of advertising opportunities and foods. The concept map will also inform the design of an evaluation of the regulations and could be used to inform the design and evaluation of similar regulations elsewhere.

## INTRODUCTION

The WHO recommends that member states limit children's exposure to marketing for less healthy foods.[1] The recommendation reflects evidence that marketing influences food preferences and consumption, both at

## STRENGTHS AND LIMITATIONS OF THIS STUDY

⇒ By including a diverse range of experts, we developed the first comprehensive articulation of the potential pathways through which new advertising regulations may impact on the commercial food system, health and society.
⇒ Holding the workshops online may have facilitated greater attendance, particularly as we employed techniques to minimise the limitations of online data collection.
⇒ Timing the workshops after sufficient details were known about the regulations allowed for a meaningful discussion about their impact but with enough time for the study's findings to feed into the regulations' design.
⇒ Though we did not aim to achieve saturation in this study, we found it difficult to recruit participants from industry.
⇒ We necessarily invited more individuals than those who ultimately participated, which may affect the transferability of the study's findings.

an individual (microlevel impacts)[2 3] and societal level (macrolevel impacts).[4] Marketing has been defined as 'the activity, set of institutions and processes for creating, communicating, delivering and exchanging offerings that have value for customers, clients, partners and society at large'.[5] Marketing is exerted through a range of activities, including those related to the product, its place, price and promotion.[6] Promotion includes building games around products (advergames), social media 'influencers' and paid for advertising in any medium. Products high in fat, salt or sugar (HFSS) are disproportionately advertised in the UK, with only 2.5% of total food and soft drink advertising spend going towards fruit and vegetables in 2020.[7] Though the causal pathways between advertising and obesity are likely to be complex,[8] it is estimated that 6.4% (95% CI: 2.0% to

## Box 1    Regulation details

It is expected that two new regulations will be implemented before the end of 2022:

1. A ban on advertisements for HFSS products shown on live broadcast TV from 05:30 to 21:00 hours ('TV advertising watershed'), including:
   a. on-demand programme services under the jurisdiction of the UK.
2. A ban on online advertisements for HFSS products, including:
   a. Non-UK regulated on-demand programme services.
   b. Social media influencers, commercial text messaging and email, all website advertising, paid-for search listings, preferential listings on price comparison sites, in-game advertisements, in-app advertising, advergames and advertorials, online display and online video.

Restrictions will not apply to 'owned media' (online property owned and controlled, usually by a brand), brand advertising, small and medium enterprises (fewer than 250 employees), audio and broadcast radio, business to business (online only) or transactional content.

'HFSS' will be defined by the 2004/2005 UK Nutrient Profiling Model and within particular categories from the Sugar Reduction Strategy. Details of the regulations may change in the lead up to implementation. Government will appoint Ofcom as the statutory regulator, who will then appoint a day-to-day regulator (expected to be Advertising Standards Authority (ASA)).[13]

13.8%) of UK childhood obesity and 5.0% (95% CI: 1.5% to 10.9%) of overweight is attributable to HFSS television advertising alone.[9]

To address concerns about the prevalence of childhood obesity, in July 2020 the UK Government Department of Health and Social Care published an intention to restrict advertisements for HFSS food and drink products on live broadcast, catch-up and on-demand television ('TV') before 21:00 hours and paid for online advertising ('online').[10] Current details of these proposed regulations are summarised in box 1, and though they have passed through the House of Lords in the Health and Care Bill,[11] details of the regulations may change before they receive Royal Assent and are implemented. Although these regulations are likely to impact on both TV and online advertising content that adults see, they have been consistently framed in policy documents as focusing on tackling childhood obesity. The first government document they were proposed in was a Childhood Obesity Strategy,[12] and subsequent strategies and policy documents have repeatedly referred to them in the context of childhood obesity.[10 13] Further, the design of the TV aspect (banning HFSS adverts from 05:30 to 21:00 hours) reflects hours when children are most likely to be watching.

The TV and online regulations proposed for the UK will be some of the most restrictive worldwide, and the first to explicitly address paid for online advertising.[14] Overall, 18% of UK advertising spend is for TV slots and at least 63% for online slots.[15] Though there has been a recent decline in broadcast TV viewing in the UK, average viewing time remains around 3 hours per day for ages 4 years and above.[15] The COVID-19 pandemic has accelerated use of subscription video-on-demand services, with viewing of services such as Netflix and Amazon Prime Video almost doubling in 2020 to an estimated 1 hour per person per day.[16] Such services would be covered by the proposed online regulation rather than the TV one. While the decline in broadcast TV viewing has been more pronounced among younger people (for 16–24 year olds down 18%, and for 4–15 year olds down 16% in 2019),[15] this has corresponded with an increase in viewing of subscription video-on-demand services among younger people (by 55 min to an average of 2 hours per day between April 2019 and April 2020).[16] It has been estimated that a pre-21:00 hours ban on HFSS TV food advertising would result in a 4.6% (1.4%–9.5%) reduction in childhood obesity and a 3.6% (1.1%–7.4%) reduction in childhood overweight prevalence.[9] Effects were twofold greater in the least compared with the most affluent social groups and would likely be amplified by comparable restrictions on online food promotion.[9] The ultimate results of such a regulation were predicted to depend on how HFSS advertising patterns change in response.[9] Though less is known about the potential effects of an online ban, emerging evidence indicates that online marketing techniques (eg, use of social media influencers) may be particularly pervasive and persuasive.[17–19]

Few evaluations of such food advertising restrictions have been conducted worldwide,[14] partly because there have been few comparable regulations. There are also challenges to evaluating this type of intervention that is delivered to whole populations and so is impractical to subject to experimental evaluation techniques such as randomised controlled trials.[20] Furthermore, the commercial food sector exhibits characteristics of a complex adaptive system.[21] Adaptations made by stakeholders residing in the system that is regulated may lead to both intended and unintended consequences that ultimately impact on the overall effectiveness of regulations.[21] The 'balloon effect' proposes that restrictions on one type of marketing can lead to increases in others,[22] as companies and other aspects of the food system adapt. Articulating these possible adaptations and their potential consequences should help refine details of the regulations before implementation. Understanding possible adaptations and consequences should also help inform the design of any evaluation.

Some other countries are following a similar path of legislation in this realm—though more often through industry self-regulation[23–27]—emphasising the need to develop generalisable evidence about the impact of the UK regulations. To maximise the applicability of evaluation findings to policymakers outside of the UK, it is helpful for evaluators to test theories as well as evaluate interventions.[28] Theory-driven evaluation first requires the development and clear articulation of programme theory.[29] Concept mapping is an approach particularly useful for public health researchers interested in developing theory.[30] A concept map is a 'diagram of proposed relationships among a set of concepts….about a particular question….or topic'.[31] Concept maps can be used

to help organise ideas, demarcate an area of interest and plan evaluations. Group concept mapping is a structured approach involving group work that is flexible to many public health contexts.[32]

## Objectives

In this study, we used an approach inspired by group concept mapping to develop a concept map of how the new TV and online advertising regulations may impact on the commercial food system, health and society. We aimed to describe how the regulations may interact with the food system so that evaluations of the regulations can be grounded in clearly articulated theory, and so that adaptations to the regulations that could improve the health impact can be identified before implementation.

## METHODS

### Study design

We created a concept map of the potential pathways through which the regulations may impact on the commercial food system, health and society. By 'food system' we mean the interdependent network of entities involved in agriculture and fisheries, food processing and production, storage and distribution, wholesaling and retailing and preparation and marketing of raw, processed and ready to eat foods.[21] By 'society', we mean the wider social system in which the food system is embedded. We developed the map using an adapted version of a group concept mapping method in four workshops.[32] The study reporting adheres to the Consolidated Criteria for Reporting Qualitative Research (online supplemental appendix 1),[33] but recognises proposed amendments relating to gender.[34]

### Participant recruitment

Workshop participants were recruited from academia, civil society, government organisations and industry (eg, food industry, media, advertising). Individuals were eligible for inclusion if they had professional knowledge and experience of food marketing regulation within their sector and were based in the UK. We identified individuals from our existing contacts in these sectors and by searching the websites of relevant organisations. In total, 63 individuals were invited by email to take part in the study (8 from academia, 15 from civil society, 11 from government organisations and 29 from industry). We aimed to recruit up to 20 individuals, approximately evenly distributed across the participant groups. As we were not aiming to reach 'saturation',[35] we decided on the number of people to recruit to the study pragmatically, based on the resources available to us but allowing for sufficient breadth.

Participants from industry attended a separate workshop to those from academia, civil society and government organisations due to the potential for conflicts of interests between sectors. We set a limit of 10 participants per workshop in addition to the facilitators (JA and HF,

who both had qualitative research experience[36 37]), which is considered a manageable total number of participants to permit dialogue and engagement.[32] Workshops were arranged around participants' availability in July and August 2021 and lasted 2 hours each.

### Data collection

Building on previous work that has used group concept mapping to inform the design of evaluations of population health interventions,[38] we used the first three steps of group concept mapping (preparation, generation and structuring)[32] and added a fourth (reflection). The first three steps were achieved in the workshops, and the final step was achieved using an online feedback form. We held the workshops on Zoom, an online videoconferencing software (https://zoom.us/), to minimise time demands on participants and as data collection took place during COVID-19 restrictions. In the workshops, we used a combination of prepiloted Microsoft PowerPoint slides and Miro software (https://miro.com/) to provide instructions to participants and visualise their contributions as they were made, respectively. Our data consisted of screenshots of maps as they developed, the map from each workshop, audio recordings of the workshops and postworkshop feedback returned through an online form. Workshops were held under the Chatham House Rule:[39] participants were told they could use the information discussed in the workshops, but they could not reveal the identity or affiliation of other participants. Figure 1 summarises the method used to develop the final concept map.

### Preparation

Preparation entailed setting out the aims and processes of the workshop and agreeing the focus area of the map.[32] At the beginning of each workshop, the workshop facilitators introduced the aims and processes. They reminded participants of the intervention details, the withdrawal process and that the workshops were being recorded. The facilitators proposed that the focus area was 'what are the potential pathways through which the intervention might impact on health, the commercial food system and society?'. Participants were invited to help refine this during a discussion of approximately 5 min.

### Generation

Generation is a divergent process where participants individually brainstorm a long list of responses to the focus area and consider the relative importance of each response.[32] Participants were given around 10 min to independently generate a list of as many responses as possible to the refined focus area, including pathways to both positive and negative impacts arising from the regulations.

### Structuring

Structuring is a convergent process where participants organise and critically reflect on ideas and relationships between concepts.[32] For approximately 60 min, participants were asked in turn to contribute responses to the

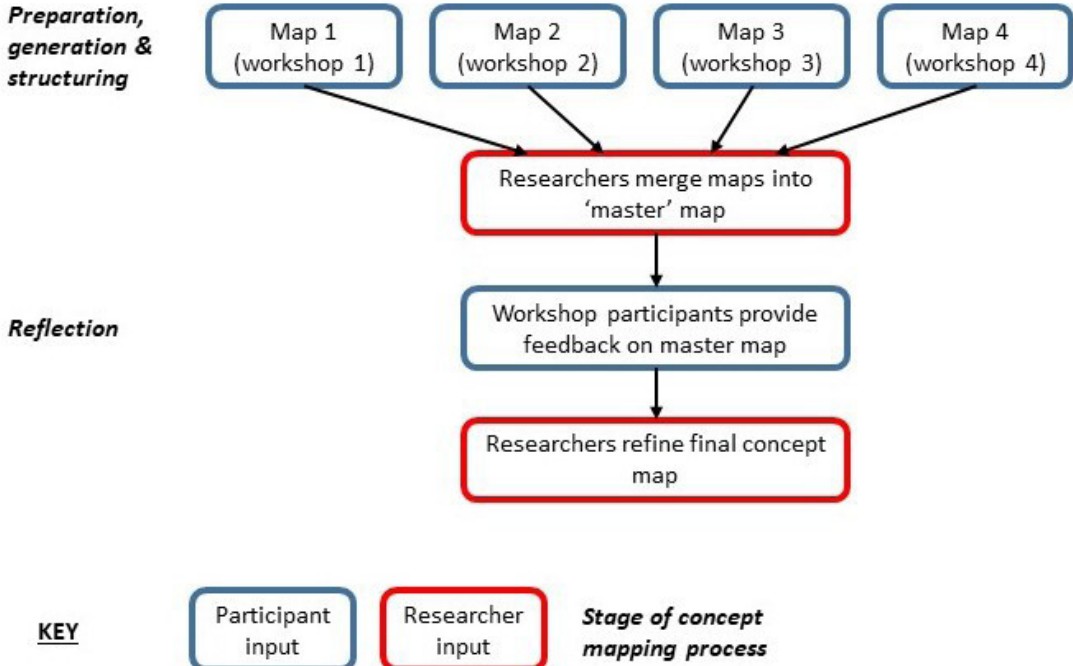

**Figure 1** Summary of method used to develop the concept map.

focus area from their individual brainstorming in order of relative importance. These were structured and visualised in real-time using Miro, which was shared on-screen with participants, with new concepts and relationships added to a draft map as participants suggested them (see figure 2). Once all responses were included, participants were invited to reflect on the map, adding additional concepts and relationships as required. We adopted an inclusive approach to adding concepts and relationships to maps, including everything mentioned and not deleting anything previously added.

### Reflection

After the workshops, we merged the map from each workshop into one 'master' map. We used a method inspired by those employed in other mapping projects.[40] First, HF charted all concepts in the maps into a Microsoft Excel sheet, and similar or identical concepts across the maps were grouped and refined into simplified concepts and accompanying descriptions. Second, these refined concepts were mapped in a way that corresponded with pathways depicted in the four separate maps. Concepts not immediately fitting anywhere were placed to the side for further deliberation with JA. As we took an inclusive approach, all concepts from the individual maps contributed to the master map. The master map was discussed with the wider research team (EJB, PS, MW, RS) and steering committee, prompting some minor changes but notably, no areas of significant disagreement.

We then circulated the master map to all workshop participants by email. The email contained a link to an online form issued via REDCap (https://www.project-redcap.org/) that asked questions about the map to seek suggested changes. We used the suggestions to produce a final concept map.

### Analysis

Beyond merging the maps from each workshop into a master map, no formal analyses were conducted.

### Patient and public involvement

Patients and/or the public were not involved in the design, conduct, reporting or dissemination plans of this research.

### RESULTS

From four workshops with a total of 14 participants, we developed a concept map to describe how the proposed TV and online advertising regulations may impact on the commercial food system, health and society. Here we present the concept map and describe its component concepts.

### Participant characteristics

We held four workshops: one with individuals from industry, and three with individuals from academia, civil society and government organisations (see table 1). As the focus was on generating the map as a group, we did not collate any demographic information about participants.[40]

### Concept map of anticipated adaptations to the regulations

The maps produced in each workshop are provided in online supplemental appendix 2, and they illustrate the nuance in focus between workshops. For example, the

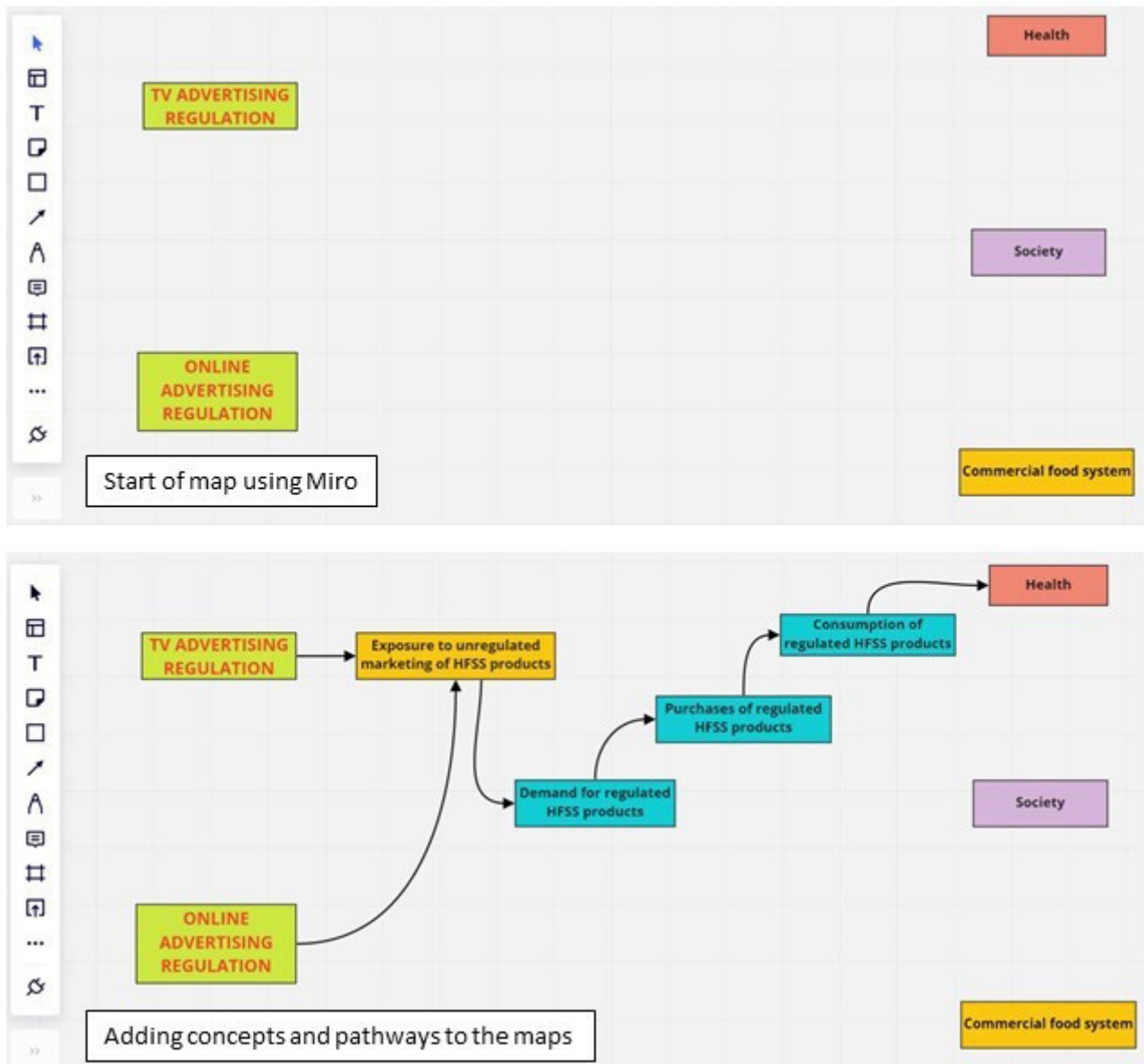

**Figure 2** Examples of mapping concepts and pathways using Miro. HFSS, high in fat, salt or sugar.

workshop with industry participants focused more on the technical difficulties presented by the regulations than in other workshops. Six workshop participants provided feedback on the master map during the reflection stage (academia=2, civil society=3, government organisation=1). In response to the feedback, we refined some of the connections between concepts (eg, adding a direct link connecting health and employment), and highlighted the regulations to make them more visibly striking.

The resultant concept map is presented in figure 3, and it depicts the possible pathways of change that could follow the regulations. Colour coding is used to differentiate the groups of reactions to the regulations: government,

**Table 1** Sectors included in each workshop

| Participant sectors per workshop | Workshop 1 | Workshop 2 | Workshop 3 | Workshop 4 | Total |
|---|---|---|---|---|---|
| Academia | 2 | 1 | 1 | 0 | 4 |
| Civil society | 2 | 1 | 3 | 0 | 6 |
| Government organisation | 0 | 1 | 1 | 0 | 2 |
| Industry | 0 | 0 | 0 | 2 | 2 |
| | | | | Grand total | 14 |

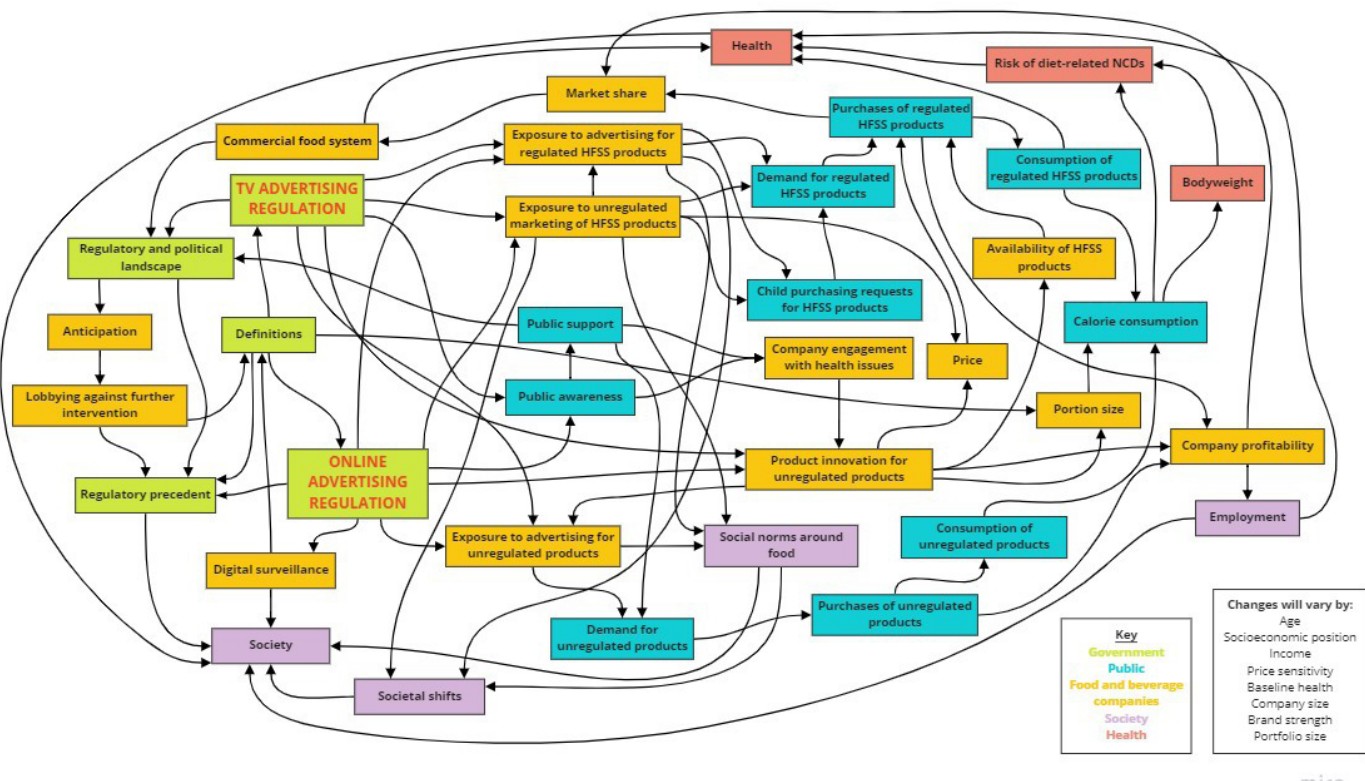

**Figure 3** Concept map of pathways through which the proposed UK TV and online advertising regulations may affect the commercial food system, health and society. HFSS, high in fat, salt or sugar.

food and beverage companies, public, society and health. Pathways depicted are not exhaustive, as it is possible that other links between concepts exist that were not captured in the workshops. The map is also accompanied by a list of factors that may modify the impact of pathways that it depicts, such as socioeconomic position and company size. The concepts contained in each workshop map, and the corresponding concepts they were assigned in the final concept map, are provided in online supplemental appendix 3. Concepts are described in more detail in table 2.

## DISCUSSION
### Overview of findings

Using an adapted group concept mapping method in four expert workshops, we developed a concept map to visualise how the proposed TV and online food advertising regulations may impact on the commercial food system, health and society. The concept map illustrates that the pathways between the regulations and these impact domains will be determined by the reactions of stakeholders.

### Strengths and limitations

To our knowledge, this is the first cross-sectoral attempt to explicitly theorise how regulations of this kind may impact on the commercial food system, health and society. Incorporating the views of a range of experts with different perspectives and interests allowed us to create

a comprehensive articulation of the ways the regulations may positively or negatively affect public health. As with any qualitative research, our map does not claim to be representative of all views, nor comprehensive, of the wider groups that participants represent.[40] Instead, we intended to sample a diverse range of expert views related to food marketing and its regulation. Including participants from diverse sectors is a strength of the study as it enabled the proposed regulations to be theorised expansively. Nonetheless, it is possible that other concepts and pathways may exist but were not captured by our map.

We necessarily invited more individuals than those who ultimately participated. The timing of the data collection period was a common reason for non-participation in the workshops, as it coincided with summer and school holidays in the UK, which may have made it difficult for those with child caring responsibilities to attend. To accommodate individuals' other commitments, we held smaller workshops across various times and days. Doing so increased the participation in our study, but it may have lost some discussion and synergy that larger groups allow.

We found it difficult to recruit individuals from industry and government organisations. Employees from these sectors rarely have their contact details listed on public-facing websites, unlike those from academia and civil society. Government organisations expressed reluctance to contribute information beyond what was already in the public domain.[41] There may have also been reluctance from industry to engage with our research due to

**Table 2** Description of concepts in the concept map

| Statement | Description |
|---|---|
| Anticipation | Food and drink companies foresee the introduction of the regulations,* and possibly other related legislation for example, volume and location price promotion. |
| Availability of HFSS products | Availability of *all* HFSS foods and beverages, both within and outside the scope of the regulations.* in physical and online shops. |
| Bodyweight | In terms of BMI, overweight or obesity status. |
| Calorie consumption | Total energy intake of individuals. |
| Child purchasing requests for HFSS products | Degree to which children make purchasing requests to caregivers for *all* HFSS products, both within and outside the scope of the regulations.* |
| Commercial food system | Interdependent networks of commercial entities involved in agriculture and fisheries, food processing and production, storage and distribution, wholesaling and retailing, and preparation and marketing of raw, processed and ready to eat foods.[21] |
| Company engagement with health issues | Degree to which food and beverage companies orientate their business around public health goals. |
| Company profitability | A company's ability to make profit. |
| Consumption of regulated HFSS products | Individual's intake of foods and beverages within the scope of the regulations.* |
| Consumption of unregulated products | Individual's intake of foods and beverages that are not within the scope of the regulations.* |
| Definitions | Information used to define or enforce the regulation,* including the UK Nutrient Profiling Model and the food categories from the Sugar Reduction Strategy. Importantly, the regulations* cover a group of foods that is different from those covered by other UK dietary public health regulations. Enforcement is based on information provided by companies. |
| Demand for regulated HFSS products | Public desire to purchase or consume foods and beverages within the scope of the regulations.* |
| Demand for unregulated products | Public desire to purchase or consume foods and beverages outside of the scope of the regulations.* |
| Digital surveillance | Digital data collated by website to inform regulation* enforcement. |
| Employment | Number of people employed in the commercial food system. |
| Exposure to advertising for unregulated products | Exposure† to adverts for products outside of the scope of the regulations. For foods and beverages, this could be HFSS products within companies' portfolios that are outside of the scope of the regulations, healthier products (eg, fruit and vegetables), or food delivery companies. Also includes non-food and beverage products and services, but not clear what health impacts they might have. |
| Exposure to advertising for regulated HFSS products | Exposure† to advertising for food and beverages within the scope of the regulations.* |
| Exposure to unregulated marketing of HFSS products | Exposure† to advertising for *all* HFSS products on media that are outside of the scope of the regulations.* Includes offline advertising (eg, print media), forms of marketing online that are exempt from the regulations (eg, in owned media), sponsorship, brand advertising and creative modes of marketing that are hard to capture with regulation. |
| Health | Overall health, including and beyond bodyweight and NCDs. |
| Lobbying against further interventions | Activities undertaken by, or on behalf of, food and beverage companies to resist further policy or regulations. |
| Market share | The size of the total market held by a company. Few companies that each hold a large market share creates a concentrated market. |
| Portion size | Size of food and beverage products in grams or calories, or recommended portion size. |
| Price | Price of food and beverage products, including price discounts. |
| Product innovation for unregulated products | Developing new products that are outside of the scope of the regulations,* or reformulating existing products so they are no longer within the scope of the regulations. Could include reformulation using artificial ingredients or developing for example, saltier products that are currently an exempt category. Some categories of products are easier to change than others, and some companies are better able to respond in this way than others. |
| Public awareness | Degree of public awareness of both the regulations* and the problems they are trying to address. |

**Table 2** Continued

| Statement | Description |
| --- | --- |
| Public support | Degree of public support for the regulations.* |
| Purchases of regulated HFSS products | Sales (from company perspective) or purchases (from individual perspective) of food and beverage products within the scope of the regulations.* |
| Purchases of unregulated products | Sales (from company perspective) or purchases (from individual perspective) of food and beverage products outside of the scope of the regulations.* |
| Regulatory and political landscape | Wider landscape of regulation and policy, including others relating to marketing (eg, location and volume price regulations) and COVID-19. The degree to which the regulations* harmonise with the wider political and regulatory landscape. |
| Regulatory precedent | Implementation of the regulations* serves as precedent for any future regulation. |
| Risk of diet-related NCDs | Risk of developing NCDs influenced by dietary behaviours. |
| Social norms around food | Implicit or explicit beliefs, attitudes, or behaviours about eating, at both an individual and family level. |
| Society | The wider social system in which the food system is embedded. |
| Societal shifts | Exposure† to advertising affects social norms and may contribute to societal changes in consumerism and culture. |

*The regulations apply to online and TV advertising for a subset of HFSS products, defined by the 2004/2005 UK Nutrient Profiling Model and within particular categories from the Sugar Reduction Strategy. This means there are HFSS products (unregulated HFSS) and non-HFSS products outside of the scope of the regulations.
†Exposure is a function of advertising prevalence, but is also dependent on individual-level factors (eg, frequency of media use).
BMI, body mass index; HFSS, high fat, salt and sugar; NCDs, non-communicable diseases.

inherent differences between the goals of public health researchers and of the food industry. Including a relatively small number of industry representatives may have limited our final map, and those industry perspectives in our study may be more sympathetic to public health goals than those of the wider sector. However, one of the representatives of industry we did include worked for an umbrella group and so may have a particularly broad perspective to bring. Some of our participants representing other sectors also had previous experience of working with industry. Participants may have also taken part in our study to pursue their own agenda, as industry actors have previously sought to undermine food advertising regulations.[42 43] There are some differences in the contributions made by industry participants compared with non-industry ones (online supplemental appendices 2 and 3). However, the nature of the workshop content, holding workshops with experts from non-industry sectors, and verifying findings with all participants, left little room for industry interests to overly dominate our concept map.

Conducting the workshops in person may have achieved different results, as some participants may have felt more able to share sensitive information in person. However, online workshops widened attendance to those who would have been unable to attend in-person. To avoid some of the potential challenges of collecting data using Zoom, we employed several recommended strategies.[44] This included using screen-sharing and clear greetings to develop rapport, using back-up recording devices, holding facilitator briefings to avoid technical issues and establishing 'house rules' to ease participants' experiences.[44] To maintain participant engagement, workshop duration was limited to 2 hours, and primarily focused on capturing concepts rather than exhaustively detailing the pathways between them. Though it may have increased participant fatigue and burden, holding longer workshops may have allowed time to capture additional concepts and pathways. As a form of member-checking,[45] we verified the master map with all workshop participants by email, in a further attempt to ensure the final concept map accurately represented participants' contributions and to allow additional comments.

### Interpretation of findings

The concept map can be used to illustrate pathways through which the reactions of food and drink companies may serve or undermine the public health goals of the regulations. Here, as previously in work using similar methods,[46] we describe three potential scenarios: (1) adaptations are made to the regulations in ways that reinforce positive impacts on public health (see figure 4); (2) adaptations are made to the regulations in ways that undermine impacts on public health (see figure 5); and (3) technicalities of the regulations cover too few unhealthy food products and advertising opportunities to make a substantial difference to public health (see figure 6). As it is unlikely all companies will respond uniformly, a combination of the three scenarios may follow the implementation of the regulations.

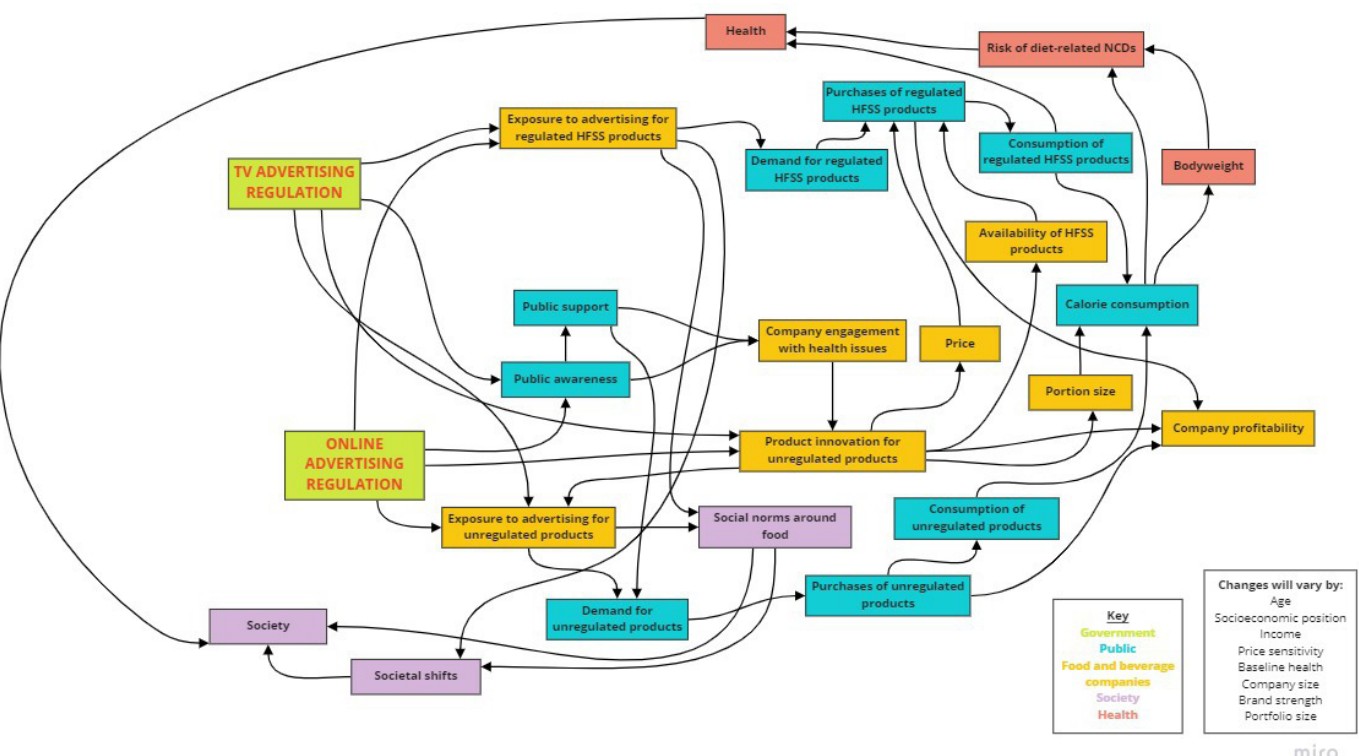

**Figure 4** Scenario 1: adaptations reinforce positive impacts of the regulations on public health. HFSS, high in fat, salt or sugar.

## Scenario 1: adaptations reinforce positive impacts of the regulations on public health

Companies may reduce their TV and online advertising for regulated HFSS products, as they will have less opportunity for advertisements. Doing so reduces people's exposure to HFSS adverts, which may prompt corresponding reductions in demand, purchases and consumption of the associated HFSS products.

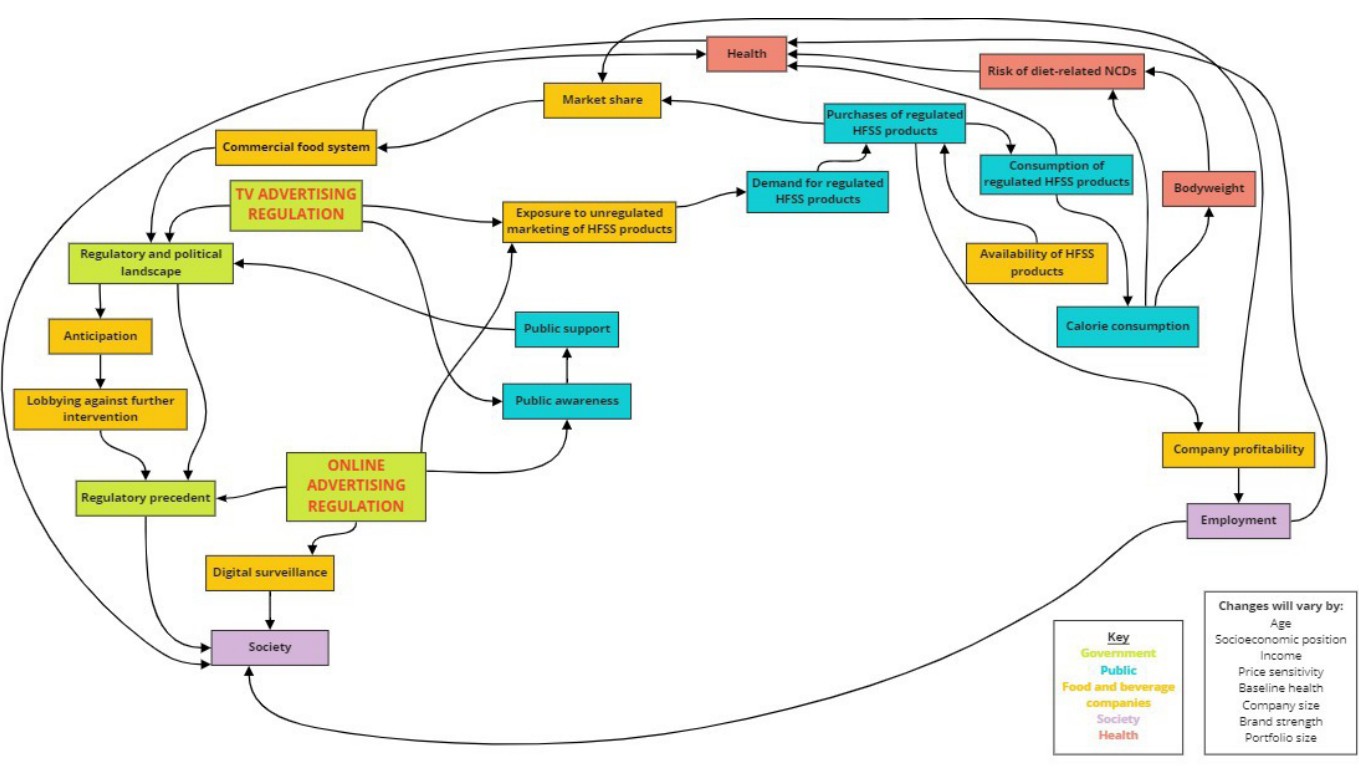

**Figure 5** Scenario 2: adaptations undermine impacts of the regulations on public health. HFSS, high in fat, salt or sugar

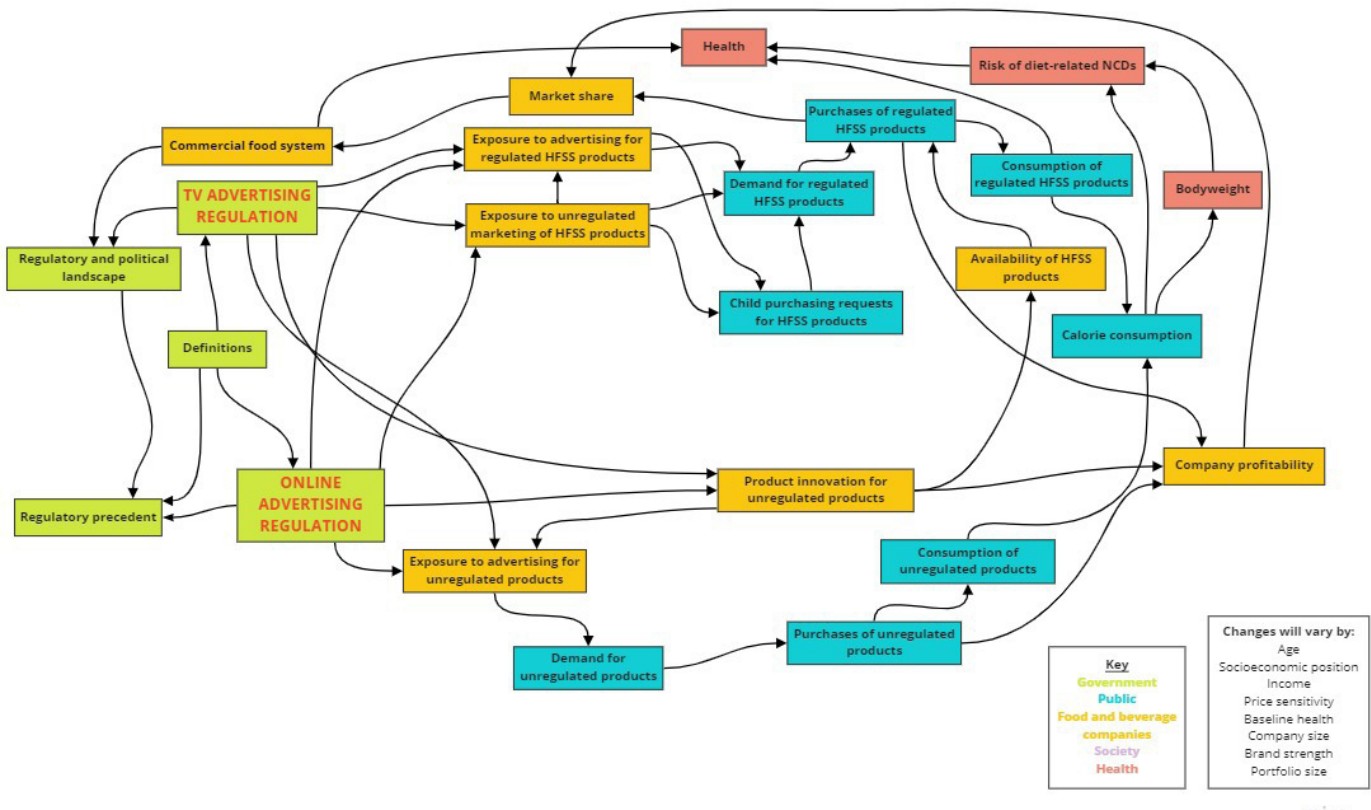

**Figure 6** Scenario 3: technicalities hinder potential impacts of the regulations on public health. HFSS, high in fat, salt or sugar; NCDs, non-communicable diseases.

Consequently, this will reduce the total number of calories consumed by individuals, improving health outcomes both associated with, and independent of, body weight.

To make up lost revenue from fewer HFSS product purchases, companies may increase TV and online advertising for their products that are out of the scope of the regulations (eg, 'spotlighting' low-fat, low-salt or low-sugar alternatives). They may also engage with diet-related health issues, which could include developing and advertising new products that are out of scope of the regulations, particularly if there is public support for the regulations and corresponding falls in demand for HFSS products. Doing so reduces the proportion of HFSS products (relative to non-HFSS) available in the food system.

Reduced exposure to HFSS adverts may change social norms about the acceptability of consuming HFSS products. It may also change a consumerism mindset that may be encouraged by adverts to over-purchase and consume products. These changes could contribute to societal shifts that reinforce lower demand for HFSS products and change macro-level eating behaviours.

### Scenario 2: adaptations undermine impacts of the regulations on public health

Food and drink companies could also minimise losses incurred by the regulations by redirecting their efforts towards unregulated forms of marketing ('balloon effect'). Companies could increase their expenditure on brand advertising, sports sponsorship or advertising outdoors or in print or audio media, none of which are intended to be covered by the regulations. In their marketing messaging, companies could also reframe diet-related health issues to position inactive lifestyles as a more substantial contribution to non-communicable diseases. It is unclear how this may affect people's total exposure to marketing, and their resultant demand for HFSS products. Companies may also fear the implementation of further regulations that could affect their performance, and so may lobby against them. Lobbying could change future regulations such that their impact is limited, and in turn, may mean that other, comparable regulations also have less chance of being implemented.

To implement regulations, companies may increase the amount of data they collect about the population. Such data gathering constitutes greater digital surveillance that impacts society (eg, privacy rights),[47] but could also inform more targeted marketing that is known to be highly effective at encouraging sales and consumption.[48–50]

### Scenario 3: technicalities hinder potential impacts of the regulations on public health

The regulations have a specific set of HFSS within scope, which has notable exemptions such as some salty foods. TV and online advertising for products exempt from the regulations may continue, as may the corresponding purchasing and consumption of these products. Some participants reported that the proposed scope of the regulations differ to that of other policies. Lack of

consistency with other regulations may make it costly—perhaps to the point of being futile—for companies to respond to the regulations by developing new products that are compliant with all related regulations. Limited development of new products would restrict the degree of transformation in the food system. Furthermore, unlike other regulations, these advertising regulations are not defined by portion size nor are smaller portion sizes an explicit objective of the regulations. This means there is no incentive for companies to produce smaller product sizes, which could otherwise contribute towards reducing calorie consumption via HFSS products.

As advertising by small and medium enterprises are also exempt from the regulations, larger companies may 'atomise' by creating smaller off-shoot companies, which can continue to advertise and sell HFSS products without limitation by the regulations. Advertising of HFSS outside of the watershed hours will still be permitted on TV and on-demand services, and large HFSS companies can afford the high price of advertising slots likely to occur after 21:00 hours. TV advertising after 21:00 hours may therefore become saturated with HFSS products, which may limit the impact of the regulations on adults' and older teenagers' consumption habits and, by extension, that of the children they are responsible for.

### Comparison to existing literature

Many existing models exist to illustrate how food marketing affects behaviour and health[8] and logic models are regularly produced to illustrate how other diet-related public health regulations may work. Methods for developing such models have evolved to appreciate the complexity of the surrounding system in which they reside,[51] but to our knowledge, these have been rarely applied in the context of diet-related health interventions,[46] and not applied to food advertising regulations before. The concept map we developed here is the first we are aware of to show how food marketing regulations may work by interacting with their surrounding system.

The concept map we developed illustrates ways that reactions to the regulations will reinforce or undermine their impact on public health, reinforcing the hypotheses of earlier work.[9] The potential for some of these pathways to exist has been evidenced elsewhere. Analyses have found that 57 of 65 brands associated with HFSS had an easily identifiable HFSS product, and the majority (84%) of these products had an alternative non-HFSS product from the same brand, master brand, parent company or license holder company brand portfolio in the UK.[52] Evidence also indicates that HFSS companies have reformulated and developed new products in responses to diet-related polices in the UK, such as the Soft Drinks Industry Levy.[53] This evidence corresponds with pathways in the map that show how companies could redistribute advertising from regulated to unregulated products.

Pathways that illustrate the risk of food companies undermining the regulations may be particularly plausible given existing evidence has documented industry opposition to HFSS advertising regulations in the UK.[42 43] The UK government's Department for Digital, Culture, Media & Sport impact assessment of the regulations also assumed that a degree of HFSS advertising will be displaced to other media,[31] as has existing research on the TV regulation specifically.[9 54] It is also widely documented in broader literature that efforts to undermine such regulations often form part of wider market strategies that, when exerted by powerful and global corporations, are difficult to address with singular regulations.[55] Our concept map builds on this evidence by elucidating pathways through which regulation may be undermined, from which it may be possible to adapt the proposed regulations or implement additional, complementary ones to maximise the likelihood of the regulations achieving their public health goals.

### Implications and further research

As the TV and online advertising regulations are not yet implemented, our findings could be used to augment the proposed legislation to encourage stakeholder reactions that maximise the regulations potential benefits. Ensuring that definitions underpinning the legislation, particularly those relating to product categories, harmonise with other legislation affecting commercial food providers may double-down the incentive to reformulate or develop new, non-HFSS products rather than market HFSS products by other means. Expanding the existing definition to a wider range of foods (eg, salty snacks currently exempt) could have the same effect. Implementing comparable regulations on other forms of marketing, such as a ban on outdoor advertising of HFSS as has been seen in London,[56] would also limit opportunity to redistribute advertising spend for HFSS. Expediting the implementation of other regulations affecting the commercial food system, such as the proposed volume and location price promotion regulations,[57] has similar potential to maximise the benefit of the TV and online advertising ones by limiting opportunities for redistributing efforts to unregulated marketing. Some of these proposed alterations echo responses to the Department of Health and Social Care, and Department For Digital, Culture, Media and Sport 2020 policy consultation.[13] That they were repeated and validated by experts in multiple related fields included in our study reinforce their potential benefit.

The concept map could be used to design a complexity-informed evaluation of the regulations. Complex explanations of intervention impacts appreciate that instead of a singular cause–effect pathway, interventions can act as stimuli that send reverberations across the system in which they reside.[58 59] Complex adaptive system methods also appreciate the role of relationships between actors contributing to a variety of processes operating at different levels and scale to produce intervention outcomes.[51] In doing so, they help avoid finding a wrong answer to important questions,[60 61] and may help measure the impact of unintended consequences alongside the outcomes that the policy sets out to achieve.[62] By explicitly

exploring the connections in a complex system, these methods may also identify novel leverage points which could be targeted by future interventions. Though the map developed in our study was not explicitly conceived in systems thinking, it has many systemic qualities (eg, emphasises the role of relationships) and correlates with other methods such as 'system mapping' that have been identified as a key component of systems-informed evaluations.[51] The concept map could be used to define focal areas for evaluative studies of both the intended and unintended consequences of the regulations or could form the basis of other systems evaluation methods. This could also help establish the relative 'strength' of each relationship.

A benefit of theory, here in the form of a concept map, is that it enables the application of findings elsewhere.[28 29] The presence of food marketing regulations in other countries[63]—although different to the ones proposed in the UK—suggests there may be political appetite to learn from the UK's experience. For example, policymakers could refer to the map to consider mechanisms and pathways that are particularly relevant to their country context, and thus important to consider in developing their legislation. Findings that emerge from an evaluation based on the map would also be particularly applicable in other countries and contexts, as the maps clarify how they are embedded with other stakeholders' adaptations following the implementation of the regulations.

## Conclusions

While the proposed UK TV and online food advertising regulations will be some of the most restrictive in the world, the concept map developed in this paper illustrates that the extent to which they improve diet-related health will ultimately be determined by stakeholder reactions in the surrounding system. The map may be used as a basis for establishing a comprehensive evaluation of the UK regulations, and to inform similar regulations elsewhere. To realise the full potential of the regulations, UK policymakers may also use the map to identify and prevent loopholes in the legislations before they are implemented.

**Contributors** EJB, PS, RS, MW and JA conceived the study and acquired funding. HF and JA developed the methodology and accompanying resources and conducted the workshops. HF collated and validated the data. HF prepared the manuscript, and the draft versions were critically reviewed by EJB, PS, RS, MW and JA. All authors approved the final manuscript. JA is responsible for the overall content as the guarantor.

**Funding** This project was funded by the National Institute for Health Research Public Health Research Programme (project number 133570). The views and opinions expressed therein are those of the authors and do not necessarily reflect those of the Public Health Research Programme, NIHR, NHS or the Department of Health. HF, MW, JA are supported by the MRC Epidemiology Unit, University of Cambridge [grant number MC/UU/00006/7].

**Competing interests** JA and MW report research grants from the Medical Research Council, the Biotechnology and Biological Sciences Research Council, the Canadian Institutes of Health Research, the Department of Health & Social Care Policy Research Units, and National Institute for Health Research during the conduct of the study. MW is a member of the Expert Advisory Group of the Food Foundation, a Community Interest Company in the UK that is leading work on food insecurity, including coordinating an ongoing independent inquiry into childhood food insecurity, led by the All-Party Parliamentary Group on Hunger and Food Poverty. MW and JA received payment to their institution from Bloomberg Philanthropies for the organisation of educational events for politicians and civil servants. HF previously worked for a market research company, which conducts research on behalf of many companies, including those from the food and drink industry. HF, JA and MW have submitted evidence to the Department of Health and Social Care, and Department For Digital, Culture, Media and Sport 2020 consultation for the regulations under study, available here: https://www.cedar.iph.cam.ac.uk/resources/evidence-submissions/#HFSSadban. RS reports grants from the Medical Research Council, National Institute for Health Research, Wellcome Trust and Food Standards Agency during the conduct of the study. EJB reports grants from Public Health England during the conduct of the study. PS reports grants from British Heart Foundation, National Institute for Health Research, Medical Research Council and Wellcome Trust during the conduct of the study. The authors declare no other conflicts of interest.

**Patient and public involvement** Patients and/or the public were not involved in the design, or conduct, or reporting or dissemination plans of this research.

**Patient consent for publication** Not applicable.

**Ethics approval** The study received favourable review from the University of Cambridge School of Humanities and Social Science Research Ethics Committee in June 2021, reference number 21.276. Participants were provided with an information sheet about the study and provided informed consent before joining a workshop using an e-consent form issued via REDCap.

**Provenance and peer review** Not commissioned; externally peer reviewed.

**Data availability statement** All data relevant to the study are included in the article or uploaded as supplementary information.

**ORCID iDs**
Hannah Forde http://orcid.org/0000-0001-7447-7264
Emma J Boyland http://orcid.org/0000-0001-8384-4994
Peter Scarborough http://orcid.org/0000-0002-2378-2944
Richard Smith http://orcid.org/0000-0003-3837-6559
Martin White http://orcid.org/0000-0002-1861-6757
Jean Adams http://orcid.org/0000-0002-5733-7830

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
