## [Reviewer comments · BMJ Open]

ARTICLE DETAILS

TITLE (PROVISIONAL)	Exploring the potential impact of the proposed UK TV and online food advertising regulations: a concept mapping study
AUTHORS	Forde, Hannah; Boyland, Emma; Scarborough, Peter; Smith, Richard; White, Martin; Adams, J

VERSION 1 – REVIEW

REVIEWER	Gloria Jiménez Marín Universidad de Sevilla, Audiovisual Communication and Advertising
REVIEW RETURNED	08-Feb-2022

GENERAL COMMENTS	Bibliography is acceptable and up to date, without knowledge of self-citation. However, some interesting references are missing and it is recommended that they be added to improve the overall text. Although it is true that this is new legislation, it is also true that other countries are following this path, such as Spain, with similar prohibitions, which can help to understand this situation of prohibition, and its consequences, in society as a whole. Thus, it is suggested that the following references be revised and inserted: - Boyland, E.J.; Harris, J. L. (2017). Regulation of food marketing to children: are statutory or industry self-governed systems effective? Public Health Nutr., 20 (5) (2017), pp. 761-764- Jiménez-Marín, Gloria, Rodrigo Elías Zambrano, Araceli Galiano-Coronil, and Rafael Ravina-Ripoll. 2020. "Food and Beverage Advertising Aimed at Spanish Children Issued through Mobile Devices: A Study from a Social Marketing and Happiness Management Perspective" International Journal of Environmental Research and Public Health 17, no. 14: 5056. https://doi.org/10.3390/ijerph17145056- Sing, Fiona, Sally Mackay, Angela Culpin, Sally Hughes, and Boyd Swinburn. 2020. "Food Advertising to Children in New Zealand: A Critical Review of the Performance of a Self-Regulatory Complaints System Using a Public Health Law Framework" Nutrients 12, no. 5: 1278. https://doi.org/10.3390/nu12051278- Youngmi Lee, Jihyun Yoon, Sang-Jin Chung, Soo-Kyung Lee, Hyogyoo Kim, Soyoun Kim, Effect of TV food advertising restriction on food environment for children in South Korea, Health Promotion International, Volume 32, Issue 1, February 2017, Pages 25–34, https://doi.org/10.1093/heapro/dat078- Paul C. Coleman, Petra Hanson, Thijs van Rens, Oyinlola Oyebode (2022). A rapid review of the evidence for children's TV and online advertisement restrictions to fight obesity, Preventive Medicine Reports, 26, 101717. https://doi.org/10.1016/j.pmedr.2022.101717
---

REVIEWER	Elina Närvänen Tampere University
-----------------	--------------------------------------

GENERAL COMMENTS

It was a pleasure to read this article, which was clearly written and on a very timely topic. I have some comments to improve the manuscript:

1) I suggest you add some more discussion on the actual impacts (detailed in the different pathways) of the regulation in the abstract, so that the reader does not have to read through the whole article to see them.

2) In the introduction and background part, you discuss tv advertising to a greater extent than online advertising, which is however a bigger problem as I see it (and also identified in the current literature as less regulated but potentially more impactful than tv advertising). I would suggest you expand a little bit on the online marketing side. For suggested references, see e.g. the following:

* Ertz M, Le Bouhart G. The Other Pandemic: A Conceptual Framework and Future Research Directions of Junk Food Marketing to Children and Childhood Obesity. *Journal of Macromarketing*. 2022;42(1):30-50. doi:10.1177/02761467211054354

* Coates, Anna E., Hardman, Charlotte A., Halford, Jason C. G., Christiansen, Paul, Boyland, Emma J. (2019), "Social Media Influencer Marketing and Children's Food Intake: A Randomized Trial," *Pediatrics*, 143 (4), e20182554.

* Murphy, Gráinne, Corcoran, Ciara, Tatlow-Golden, Mimi, Boyland, Emma, Rooney, Brendan (2020), "See, Like, Share, Remember: Adolescents' Responses to Unhealthy-, Healthy-and Non-Food Advertising in Social Media," *International Journal of Environmental Research and Public Health*, 17 (7), 2181.

3) The method section is very thoroughly described. One thing that could be further specified is whether and how many responses were received at the "reflection" stage from the participants. Also, regarding the creation of the "master map" – who were involved in doing the map (more than one researcher? all authors of the paper? separately or together?) and were there any disagreements between what to include/exclude? How were these potential disagreements solved?

4) Figure 3 (Concept map of pathways) is not really clear to read, even though it nicely portrays the complexity of the phenomenon. The different pathways separately for each stakeholder that will be impacted could be outlined in addition to this figure (or at least numbered?). Perhaps depict your scenarios against the concept map? Also, while the colours depict the different stakeholders, the yellow colour (for food and beverage companies) seems to also depict influences on the public (e.g. exposure to advertising) and not only companies?

5) There is a lot of discussion and previous research specifically related to protecting children in this context. However, the UK regulations are not intended to be age-based. Thus, it could be useful to add some more discussion on this aspect. For instance, if the regulations were only directed at protecting children, industry could be more willing to accept them (as there already are voluntary industry agreements like the EU Pledge), but then on the other hand, it is very difficult to restrict online marketing based on age (because digital algorithms ultimately decide who views the ad based on user data, which is not always accurate)

6) There are some influences related to the industry that I can think of which are not covered in the current map. E.g. related to corporate social responsibility => regulations may lead to more normative pressure for food and drink companies to comply and

	decrease advertising of HFSS products in all channels even though they are not covered in the regulations (for instance outdoor advertising could easily look bad for a company's reputation even if it was basically legal to do so). Furthermore, the monitoring of the regulations is not very much addressed in the concept map. It may be a small problem that you only had 2 people from the industry to participate in the workshops and so the industry's viewpoints are not perhaps fully covered? Perhaps you can use my references mentioned above to support this perspective. Good luck with your research!
--	---

REVIEWER	Secil Deren van Het Hof Akdeniz Universitesi
REVIEW RETURNED	24-Feb-2022

GENERAL COMMENTS	The research is very interesting and it is brilliantly presented. The researchers declare that they did not aim for saturation, instead they preferred to work with 20 participants. But in the end, they have 14 participants. Although 12 participants from the civil society, government and academia can be reasonable, the food industry is represented only with 2 participants. The researchers mentioned the difficulties to recruit participants from the industry; still, only 2 participants sticks out as a weakness in such a well-designed research. Conceptualization of the members of the industry is very important since they are the most influential actors in the food system. Consequently, the number of participants from the industry needs to be increased and the maps need to be revised accordingly. Scenarios do not seem to fit in the flow of the article. First, they do not seem to come out from the workshops, thus they give the impression that they are just the subjective inferences of the researchers. Second, they do not sufficiently reflect the industrial reactions. The industry actually goes for more varied options than described in the scenarios. Their reaction to the regulations are not just limited to adaptations in their marketing policies. HFSS companies not only market new "diet" or "healthy" alternatives, but some companies also frame the issue from the perspective of "active life style", claiming that obesity is not related to the food they produce and sell, but to the inactiveness of individuals. I recommend the scenarios to be taken out from the article as a whole. Finally, concept maps usually calculate a centrality value for the concepts that emerge and also mention the strength of the relationship between two concepts either by the thickness of the line or by just putting numbers on the lines. Such visualisations add to the value of the findings. The maps can be revised accordingly as well.
---

VERSION 1 – AUTHOR RESPONSE

bmjopen-2021-060302 - "Exploring the potential impact of the proposed UK TV and online food advertising regulations: a concept mapping study"

Thank you to all reviewers and editor(s) for taking the time to provide thoughtful comments. Please note that our referencing software would not allow us to track-change our updates to the references – we have updated the references, but these updates are not marked.

Response to Editor(s)' Comments to the Authors:

*Please rework the 'Results' and 'Conclusions' sections of the abstract to include more detailed findings in the 'Results' section and to include a concise conclusion, which does not go beyond the findings/data presented in the 'Results' section, in the 'Conclusion'.

We have made these amendments to the abstract.

*You indicate that "We necessarily invited more individuals than those who ultimately participated" – please add details of this to the main text Methods section (ie, how many were invited versus agreed [response rate(s)]?).

We have added these details to the first paragraph in the 'Participant recruitment' section.

*Instead of supplying Box 1 as a separate file, please include this in the main document.

Box 1 is now included in the main document.

*Please re-upload the COREQ checklist appendix file (appendix 1) as a 'supplemental material' file, to ensure that it is available to be included as part of the final version of the paper (ie, part of the publication), rather than as a research checklist. Additionally, as the page numbers will not be accurate in the final version of the manuscript, you may wish to update this checklist to indicate sections of the manuscript rather than page numbers.

Thank you for correcting this, we have re-uploaded the checklist as supplemental material and it now refers to sections of the manuscript rather than page numbers.

*Please update the 'competing interests' statement to distinguish between funding for the present study and unrelated funding (funding 'during the conduct of the study' is unclear in this regard).

We believe we have made this distinction by including relevant funding to this study in the funding statement, and any other funding in the competing interest statement. Do let us know if you require any further clarification.

** **

Response to Reviewer 1

Dr. Gloria Jiménez Marín, Universidad de Sevilla

Comments to the Author:

Bibliography is acceptable and up to date, without knowledge of self-citation. However, some interesting references are missing and it is recommended that they be added to improve the overall text. Although it is true that this is new legislation, it is also true that other countries are following this path, such as Spain, with similar prohibitions, which can help to understand this situation of prohibition, and its consequences, in society as a whole. Thus, it is suggested that the following references be revised and inserted:

- Boyland, E.J.; Harris, J. L. (2017). Regulation of food marketing to children: are statutory or industry self-governed systems effective? *Public Health Nutr.*, 20 (5) (2017), pp. 761-764

- Jiménez-Marín, Gloria, Rodrigo Elías Zambrano, Araceli Galiano-Coronil, and Rafael Ravina-Ripoll. 2020. "Food and Beverage Advertising Aimed at Spanish Children Issued through Mobile Devices: A Study from a Social Marketing and Happiness Management Perspective" *International Journal of Environmental Research and Public Health* 17, no. 14: 5056. <https://doi.org/10.3390/ijerph17145056>

- Sing, Fiona, Sally Mackay, Angela Culpin, Sally Hughes, and Boyd Swinburn. 2020. "Food Advertising to Children in New Zealand: A Critical Review of the Performance of a Self-Regulatory

Complaints System Using a Public Health Law Framework" *Nutrients* 12, no. 5: 1278.
<https://doi.org/10.3390/nu12051278>

- Youngmi Lee, Jihyun Yoon, Sang-Jin Chung, Soo-Kyung Lee, Hyogyoo Kim, Soyoung Kim, Effect of TV food advertising restriction on food environment for children in South Korea, *Health Promotion International*, Volume 32, Issue 1, February 2017, Pages 25–34,
<https://doi.org/10.1093/heapro/dat078>

- Paul C. Coleman, Petra Hanson, Thijs van Rens, Oyinlola Oyebode (2022). A rapid review of the evidence for children's TV and online advertisement restrictions to fight obesity, *Preventive Medicine Reports*, 26, 101717. <https://doi.org/10.1016/j.pmedr.2022.101717>

Thank you for recommending these relevant publications, we have added them in appropriate places to the introduction section.

Response to Reviewer 2

Prof. Elina Närvänen, Tampere University

Comments to the Author:

It was a pleasure to read this article, which was clearly written and on a very timely topic. I have some comments to improve the manuscript:

1) I suggest you add some more discussion on the actual impacts (detailed in the different pathways) of the regulation in the abstract, so that the reader does not have to read through the whole article to see them.

We have now revised the abstract to summarise the potential impacts (though note that we still don't know what the actual impacts will be until the regulations are implemented).

2) In the introduction and background part, you discuss tv advertising to a greater extent than online advertising, which is however a bigger problem as I see it (and also identified in the current literature as less regulated but potentially more impactful than tv advertising). I would suggest you expand a little bit on the online marketing side. For suggested references, see e.g. the following:

* Ertz M, Le Bouhart G. The Other Pandemic: A Conceptual Framework and Future Research Directions of Junk Food Marketing to Children and Childhood Obesity. *Journal of Macromarketing*. 2022;42(1):30-50. doi:10.1177/02761467211054354

* Coates, Anna E., Hardman, Charlotte A., Halford, Jason C. G., Christiansen, Paul, Boyland, Emma J. (2019), "Social Media Influencer Marketing and Children's Food Intake: A Randomized Trial," *Pediatrics*, 143 (4), e20182554.

* Murphy, Gráinne, Corcoran, Ciara, Tatlow-Golden, Mimi, Boyland, Emma, Rooney, Brendan (2020), "See, Like, Share, Remember: Adolescents' Responses to Unhealthy-, Healthy-and Non-Food Advertising in Social Media," *International Journal of Environmental Research and Public Health*, 17 (7), 2181.

We agree that online marketing is potentially more impactful than television advertising, and we have updated the text with your suggested references to reflect this.

3) The method section is very thoroughly described. One thing that could be further specified is whether and how many responses were received at the "reflection" stage from the participants. Also, regarding the creation of the "master map" – who were involved in doing the map (more than one

researcher? all authors of the paper? separately or together?) and were there any disagreements between what to include/exclude? How were these potential disagreements solved?

In the methods section, we have added further details about how the authors created the master map. Please note that Appendix 3 also contains the table which was used to translate the concepts from each workshop map into the concepts included in the master map. Six workshop participants provided feedback on the final draft 'master' map, which prompted minor amendments – we have included further details of these amendments in the results section.

4) Figure 3 (Concept map of pathways) is not really clear to read, even though it nicely portrays the complexity of the phenomenon. The different pathways separately for each stakeholder that will be impacted could be outlined in addition to this figure (or at least numbered?). Perhaps depict your scenarios against the concept map? Also, while the colours depict the different stakeholders, the yellow colour (for food and beverage companies) seems to also depict influences on the public (e.g. exposure to advertising) and not only companies?

Thanks for these additional suggestions, whilst we don't think it would be helpful to add additional complexity to fig 3 we do like the suggestions of pulling out illustrations for each scenario in the discussion and have added these.

Regarding exposure, here we conceive it to be a function of marketing prevalence, and thus have suggested it is a component of company activity. However, we accept that it also depends a little on the individual e.g., how much one watches TV/goes on particular websites. We have adapted our definition in Table 2 to reflect your comment.

5) There is a lot of discussion and previous research specifically related to protecting children in this context. However, the UK regulations are not intended to be age-based. Thus, it could be useful to add some more discussion on this aspect. For instance, if the regulations were only directed at protecting children, industry could be more willing to accept them (as there already are voluntary industry agreements like the EU Pledge), but then on the other hand, it is very difficult to restrict online marketing based on age (because digital algorithms ultimately decide who views the ad based on user data, which is not always accurate)

This is a very interesting comment, and indeed something that was debated among our participants! We have carefully reviewed policy documents published since 2018 when these proposals were first made by government. In all cases, the regulations are framed in terms of preventing childhood obesity. Indeed the design of the TV bans (prevention HFSS adverts from 0530-2100hr) reflect when children are likely to be watching and when they tend to go to bed. Never-the-less it is, of course, true that these fairly comprehensive restrictions could impact on adults' exposure to HFSS marketing too. We have clarified this in the introduction.

6) There are some influences related to the industry that I can think of which are not covered in the current map. E.g. related to corporate social responsibility => regulations may lead to more normative pressure for food and drink companies to comply and decrease advertising of HFSS products in all channels even though they are not covered in the regulations (for instance outdoor advertising could easily look bad for a company's reputation even if it was basically legal to do so). Furthermore, the monitoring of the regulations is not very much addressed in the concept map. It may be a small problem that you only had 2 people from the industry to participate in the workshops and so the industry's viewpoints are not perhaps fully covered? Perhaps you can use my references mentioned above to support this perspective.

It is indeed possible that there are concepts omitted from the map. The nature of our method means we are dependent on the concepts developed by participants, and we did not use additional data

sources to triangulate this. We have added a comment to the limitations section to clarify that we cannot be sure that the map is comprehensive and there may be concepts and pathways missing.

Good luck with your research!

Many thanks!

Response to Reviewer 3

Dr. Secil Deren van Het Hof, Akdeniz Universitesi

Comments to the Author:

The research is very interesting and it is brilliantly presented.

The researchers declare that they did not aim for saturation, instead they preferred to work with 20 participants. But in the end, they have 14 participants. Although 12 participants from the civil society, government and academia can be reasonable, the food industry is represented only with 2 participants. The researchers mentioned the difficulties to recruit participants from the industry; still, only 2 participants sticks out as a weakness in such a well-designed research. Conceptualization of the members of the industry is very important since they are the most influential actors in the food system. Consequently, the number of participants from the industry needs to be increased and the maps need to be revised accordingly.

We agree that it would have benefited the research to have included more participants from industry. To this end, we spent considerably more effort trying to recruit individuals from industry than from other sectors (details of which are now included in the participant recruitment section of the main manuscript). We have added further details to the discussion about the limitations of our resultant sample. Unfortunately, we do not now have resources or ethical approval to collect additional data.

Scenarios do not seem to fit in the flow of the article. First, they do not seem to come out from the workshops, thus they give the impression that they are just the subjective inferences of the researchers. Second, they do not sufficiently reflect the industrial reactions. The industry actually goes for more varied options than described in the scenarios. Their reaction to the regulations are not just limited to adaptations in their marketing policies. HFSS companies not only market new "diet" or "healthy" alternatives, but some companies also frame the issue from the perspective of "active life style", claiming that obesity is not related to the food they produce and sell, but to the inactiveness of individuals. I recommend the scenarios to be taken out from the article as a whole.

We thought carefully about the use of scenarios and concluded that they would be valuable to include in the discussion (rather than results) section as we think that they provide useful interpretation. Other reviewers have not requested their removal. Nonetheless, we have clarified in the text that the scenarios are only illustrative to show how the map could be used, not exhaustive of all possible things that might happen. We have modified the scenarios to include your suggestions.

Finally, the examples you have given such as framing the issue as a problem of inactive lifestyle, would be part of a broad range of marketing/non-market activities that companies could use to respond to the regulations and we believe they are captured by existing concepts in the map (e.g., this example would be encompassed by the concept "company engagement with health issues"). Also note that as mentioned in response to another reviewer's comment, we have clarified that the map may not exhaustively list all possible responses to the regulations.

Finally, concept maps usually calculate a centrality value for the concepts that emerge and also mention the strength of the relationship between two concepts either by the thickness of the line or by

just putting numbers on the lines. Such visualisations add to the value of the findings. The maps can be revised accordingly as well.

Some concept maps do entail assessment of the strength of relationships between concepts either by reference to participants' perceived priority of, or certainty in, each relationship or by reference to other evidence. As described in the methods, we captured participants' perceived prioritisation of relationships by asking them to add concepts and links in order of perceived relative importance. This meant that if we ran out of time in workshops, the least important relationships were omitted. We do not believe that further quantification of relationships at this stage would be feasible. However, future work could certainly start to explore the relative strength of associations between concepts, and we have added reference to this in the discussion.

VERSION 2 – REVIEW

REVIEWER	Elina Närvänen Tampere University
REVIEW RETURNED	24-Apr-2022
GENERAL COMMENTS	Thank you for a careful revision. I have just one technical remark: - In your box 1 about the regulation details, it should probably read "small and medium companies" rather than "small and media". I also think that the low number of industry representatives is a weakness of the study, but it does not make it unpublishable, and it is sufficiently discussed in the paper.